# OpenReview forum: "Watermarking Degrades Alignment in Language Models: Analysis and Mitigation"
_TMLR — Accepted by TMLR_

### Review · Reviewer_96PR · 2025-11-27

**Summary Of Contributions:**

The paper is concerned with the loss of alignment behavior in watermarked models. Specifically, the authors claim that watermarking of a language model leads to loss in alignment behavior of the model independent of general quality loss. They claim that the loss in alignment behavior leads to lower safety guardrails in some models and extremely conservative guard-railed behavior in others. To mitigate aforementioned alignment-loss, they present an algorithm -- for a given model they generate n samples, use a reward model to rate all completions and pick the best completion. They call this Alignment Resampling (or AR). Their results show that this sampling methodology recovers the loss in alignment from watermarking, and sometimes exceeds the original model. Furthermore, this gain does not impact the detectability of the watermark significantly. Additionally, they provide a theoretical lower bound relating the gain in alignment to the number of samples required. They recognize that this method, due to its requirement of diverse samples, cannot be applied well on gumble watermarking and hence in the appendix of the paper present a modification of the gumble algorithm to add some uncertainty back.


### Key Strengths:
1. The authors do a good job in contextualizing the different tradeoffs that come with watermarking models.
2. The method presented in the paper seems to improve generations from watermarked models while preserving the watermark.
3. Empirically, it seems that the algorithm can work with samples as low as 2 or 4

### Required Clarifications and Weaknesses
1. They don't provide the metrics for when the AR algorithm is applied to the un-watermarked model. This is a crucial baseline that should be easy to run.
2. It is unclear how "overrefusal" is evaluated i.e. when is a completion considered an overrefusal? How is it differentiated from correct refusal? Please provide a clear specification of the evaluation used.
3. The reward model `R` used for the experiments and the results for the best-of-n algorithm has not been specified, only the evaluation reward model has been specified. This is an important detail that provides essential context to the reported numbers.
4. 2 out of 4 main contributions of the paper are in the appendix. If they are essential to the paper, please put them in the main paper or clarify that the paper stands without them.

**Additional Comments:**

I would be happy to reconsider my review if some of the clarifications are provided.

**Audience:**

Yes

**Audience Explanation:**

This work might be interesting to alignment researchers in speculating about the effects of watermarking on alignment.

**Broader Impact Concerns:**

No concerns.

**Claims And Evidence:**

No

**Claims Explanation:**

As per my understanding, their claims incur the following burdens:
1. They need to show that the watermark is preserved in the samples generated after AR.
2. They need to show consistent gain in alignment metrics as a result of the algorithm.
3. They need to show an improvement beyond choosing best-of-n perplexity. To elaborate, if we replace the reward model `R` with a perplexity calculation and run AR, the results should be significantly worse.
4. The theorem needs to be proved under clear and explicit assumptions.
5. Empirical validation to the theorem.

[1] **Satisfied**

[2] While their numbers suggest that this is satisfied, the number *are not enough unless they specify the reward model used*. Hence, I will mark this **unsatisfied** until the specification for `R` has been provided.


[3] **Unsatisfied:** They don't benchmark best-of-n perplexity on the metrics they evaluate the AR algorithm on. However, They benchmark best-of-n perplexity on the reward score, and report an improvement from the base model. This indicates that best-of-n perplexity for AR sampling might be very close to default AR. Hence, these numbers are required to clear that up.


[4] **Requires  Clarification**
The theorem assumes that the rewards r are gaussian with prameter $\sigma^2$ for all the policies. This shouldn't be possible because we have multiple distinct policies here (default model and watermarked model), if the difference is the mean, they should mention it. Please specify the distributions you assume each reward (for each policy) is derived from.

[5] **Unclear** : Their empirical curves in Figure 6b show the same scaling curve as suggested by the lower bound. However, the theoretical curves are always above the empirical curves (while they should probably be below).

**Requested Changes:**

## Essential Changes
Changes and clarifications that will influence my decision on this paper.
1. Specify the reward model `R` used in the experiments, and experiment with multiple reward models.
2. Provide a clear and specific description of how "overrefusals" are defined and evaluated for the results in the paper and how they are differentiated from safe refusals.
3. Benchmark best-of-n perplexity against AR on the metrics AR has been evaluated on.
4. Provide the numbers for when AR sampling is applied directly on the unwatermarked model.
5. State assumptions clearly for the theorem and the proof. Specifically, the distributional assumptions on the reward functions.
## Improvements
1. Please provide error bars.

---

> ### Author Response · Authors · 2025-12-13
> **Response to Review (Part I)**
>
> We thank the reviewer for their careful critique of our work. We are encouraged that the reviewer has found our proposed Alignment Resampling procedure as a promising mitigation tool against watermark-induced alignment degradation. The reviewer's articulation of "burdens" associated with our claims was very helpful in clarifying what must be demonstrated in the paper to be convincing. It has helped us to conduct additional experiments to refine the draft, strengthening both our empirical evidence and theoretical framing. We have updated the pdf with additional results and changes which are marked in a different color which we plan to remove upon acceptance.
>
> We address the comments as follows
>
> **1. Specify the reward model `R` used in the experiments, and experiment with multiple reward models.**
> We apologize for this oversight. The reward model used in our experiments was the Armo Reward Model (ArmoRM (https://arxiv.org/abs/2406.12845).  We add a clarifying line specifying the reward model used in the main text of the paper
>
> In addition, we experimented with a SkyworkRM as a second reward model (https://arxiv.org/abs/2410.18451). We choose this model since this is a competitive model on the reward bench leaderboard (https://huggingface.co/spaces/allenai/reward-bench). When we were writing the paper ArmoRM was among the top 10 on the leaderboard. These additional results are included in a new section in the appendix titled `Appendix K: Generalization to Alternative Reward Models`
> The claims hold with the SkyworkRM suggesting that our proposed procedure generalizes across reward models. Furthermore, to rule out the issue of dataset contamination, as pointed by another reviewer, we also carefully collected the list of training datasets used in the ArmoRM paper and carefully compared them against our datasets. We don't see any test set contamination confounding our results. To clarify this we add the following analysis in `Appendix: K Generalization to Alternative Reward Models`
>
>
> **2. Provide a clear and specific description of how "overrefusals" are defined and evaluated for the results in the paper and how they are differentiated from safe refusals.**
> We study overrefusals using two existing benchmarks from the literature (XSTest and OR-Bench). These contain hard negative examples i.e. examples which are safe but might trip a LM to go into an overcautious mode due to the choice of wording. These are different from safe refusals i.e. refusals to legitimately harmful queries which are captured in the safety assessment.
> To clarify this, we add the clarifying text in overrefusal assessment section. We also updated Figure 1 in the paper to aid the explanation through an illustrative example.
>
> **3. Benchmark best-of-n perplexity against AR on the metrics AR has been evaluated on.**
> We thank the reviewer for raising this important point. We agree that our original submission did not fully discharge the evidentiary burden associated with comparing Best of N with Perplexity Against AR. The reviewer is correct that our claim needed further empirical support. To supplement the reward score results, we conducted a new set of experiments and report the exact truthfulness, safety and overrefusal metrics; similar to what has been reported in the plots in the main section of the paper for the four models in `Appendix I: Best-of-N using Perplexity: Why Standard Quality Metrics Fail for Alignment.` The results of these experiments show that best-of-n using perplexity does not consistently recover truthfulness, safety and overrrefusal across the various models tested. This additional datapoint directly addresses the reviewer’s concern and substantiates our claim that perplexity-based selection is not an adequate alternative to AR for alignment restoration.
>
> **4. Provide the numbers for when AR sampling is applied directly on the unwatermarked model.**
>
> We thank the reviewer for raising the question of applying AR directly to the unwatermarked model. Our primary objective, however, is not to improve the unwatermarked model per se, but to restore alignment behavior in **watermarked** generations without sacrificing watermark detectability. We agree that, as a generic best-of-_n_ procedure with an alignment-targeted reward, AR applied directly to the unwatermarked model would be expected to further improve its alignment metrics. For this reason, we did not treat AR-on-unwatermarked as a central baseline in the original draft: it does not inform the key deployment trade-off between watermarking and alignment, which is the focus of our study. We hope this clarifies our reasoning. Please let us know if we can answer any further questions on this concern.

---

> > ### Author Response · Authors · 2025-12-13
> > **Response to Review (Part II)**
> >
> > **5. State assumptions clearly for the theorem and the proof. Specifically, the distributional assumptions on the reward functions.**
> > We thank the reviewer for requesting this clarification. The reward distributions under both policies are indeed Gaussian with identical variance $\sigma^2$ but different means. Specifically, we assume $r|\pi_{ref} \sim N(\mu_{ref}, \sigma^2)$ and $r|\pi_w \sim N(\mu_w, \sigma^2)$ where $\mu_{ref} - \mu_w = ε$ represents the watermarking-induced degradation. The shared variance assumption is reasonable because watermarking primarily shifts the reward distribution's location (by biasing toward certain tokens) rather than fundamentally altering its spread. We will clarify this in the revised manuscript by explicitly stating: `We assume rewards follow Gaussian distributions with shared variance σ² across policies, differing only in their means by the degradation term` $\varepsilon$.
> >
> > **However, the theoretical curves are always above the empirical curves (while they should probably be below).**
> > We appreciate the reviewer’s point that a genuine lower bound should, by definition, remain below the empirical curves. The derivation of our bound relies on several simplifying assumptions, including independence of samples, Gaussianity of the reward scores, and a plug-in (empirical) estimate of the variance. hese assumptions such as independence, Gaussian reward distributions, and the use of an empirical variance estimate—introduce approximation error relative to the finite-sample, non-idealized setting of our experiments. Consequently, the resulting expression should be interpreted as an approximate scaling law rather than a numerically tight lower bound, and a nontrivial theory–practice gap is to be expected. Taken together, these modeling choices inevitably create a mismatch between the “idealized” theoretical prediction and the empirical curves, which can manifest as occasional crossings. In particular, across different temperatures, there are regimes where the empirical (solid) curves lie above the theoretical (dotted) ones, consistent with the lower-bound interpretation, as well as regimes where the reverse occurs due to approximation error. We will add a sentence to make this clearer in the main text.
> > `Because the bound is derived under idealized assumptions (independent, approximately Gaussian rewards with empirically estimated variance), it serves primarily as an approximate scaling guide, and some mismatch with the empirical curves, including occasional crossings, is to be expected.`

---

> ### Author Response · Authors · 2025-12-13
> **Response to Review (Part III)**
>
> In addition, we seek to provide clarification on some of the points raised above
>
>
> **As per my understanding, their claims incur the following burdens:**
>
> 1. **They need to show that the watermark is preserved in the samples generated after AR.**
>    We agree this is essential. As shown in Table 1 (page 11), watermark detectability (F1) remains high and very similar with and without AR, indicating that AR restores alignment while largely preserving watermark strength.
>
> 2. **They need to show consistent gain in alignment metrics as a result of the algorithm.**
>    As detailed above, we have added experiments that explicitly report truthfulness, safety, and overrefusal under AR, mirroring the main-paper plots. These results confirm that AR consistently improves alignment metrics over the watermarked baseline across the models we study.
>
> 3. **They need to show an improvement beyond choosing best-of-n perplexity.**
>    We ran new experiments replacing the reward model $R$ with perplexity and evaluating truthfulness, safety, and overrefusal. As reported in `Appendix I: Best-of-N using Perplexity: Why Standard Quality Metrics Fail for Alignment`, best-of-$n$ perplexity does *not* consistently recover alignment, whereas AR does, showing that an alignment-aware reward outperforms perplexity-based selection.
>
> 4. **The theorem needs to be proved under clear and explicit assumptions.**
>    We have revised the theory section to explicitly state the distributional assumptions:
>    $r \mid \pi_{\text{ref}} \sim \mathcal{N}(\mu_{\text{ref}}, \sigma^2)$ and
>    $r \mid \pi_w \sim \mathcal{N}(\mu_w, \sigma^2)$,
>    with $\mu_{\text{ref}} - \mu_w = \varepsilon$ capturing watermark-induced degradation and shared variance $\sigma^2$ reflecting a location shift rather than a change in spread.
>
> 5. **Empirical validation to the theorem.**
>    Our empirical curves (e.g., Figure 6b and the additional temperature plots) are meant to validate the *scaling behavior* predicted by the theorem. We clarify that, due to idealized assumptions (independence, approximate Gaussianity, empirical variance), the bound serves as an approximate scaling guide, so a visible theory–practice gap and occasional crossings with the empirical curves are expected.
>
>
> Thank you again for your thoughtful engagement with our work. Your questions have helped us improve the presentation of the paper by clarifying some of the choices that we have made while designing and conducting this study. Please let us know if our response addresses the questions that you have raised.

---

### Review · Reviewer_9eYk · 2025-11-27

**Summary Of Contributions:**

The paper analyses the 2 main techniques for watermarking the output of Large Language models, and shows how they lead to a weakening of the effects of alignment techniques. The authors also look at different ways these techniques lead to degradation in safety, truthfulness, and helpfulness.
Finally the paper also presents a technique based on best-of-n sampling that effectively recovers the effects of model alignment, i.e. recovers their original safety, truthfulness, and helpfulness, while still maintaining the watermarking detectability

**Audience:**

Yes

**Audience Explanation:**

Large Language Models are becoming more capable and more widely used every day, and research on them has grown exponentially over the past few years. Alignment and Watermarking are 2 sub-areas of Language Model research with clear applications to safety and to our ability to even distinguish what's written by a human as opposed to generated by an LLM.
This paper shows that the techniques from these fields end up clashing with one another, and proposes a method that allows us to have aligned models whose generations are detectable, which has significant theoretical and practical impact.

**Broader Impact Concerns:**

I believe the impact of the paper is broadly postive

**Claims And Evidence:**

Yes

**Claims Explanation:**

The paper presents many empirical results both on the degradation of model aligment caused by watermarking, but also on how best-of-n sampling can mitigate or completely nullify such degradation. Furthermore the authors also provide theoretical justification for the phenomena, and some intuitive causal mechanisms for why watermarking techniques could harm alignment in language models.

**Requested Changes:**

No requested changes.

---

> ### Author Response · Authors · 2025-12-13
> **Thank you for the review**
>
> Thank you for taking the time to review our submission and for providing such thoughtful and encouraging feedback. We sincerely appreciate your careful assessment of our empirical and theoretical contributions, as well as your perspective on the relevance of this work to the TMLR community. Your comments reinforce the importance of understanding the interaction between alignment and watermarking methods, and we are grateful that you found the results clear, well-supported, and of potential practical value. Thank you again for your time and constructive evaluation.

---

### Review · Reviewer_n8UR · 2025-12-01

**Summary Of Contributions:**

The authors provide a study regarding changes observed after watermarking instruction-tuned LLMs. The changes observed by the authors are that post-watermarking the LLMs result in: 1) underperformance on benchmarks measuring model safety alignment, like truthfulness, safety, etc., and 2) increased overrefusal rates and unsafe responses. Multiple LLMs are evaluated on these specific benchmarks by the authors with convincing results (Fig-3,4).

Next, the authors analyze "why" these behaviors occur on watermarked LLMs. The general idea is that the watermarking objective is in some way orthogonal to the alignment objective. Some words (tokens) that promote high alignment are usually steered based on the first few token generations. Watermarking adds a constraint on the randomness of tokens, which inhibits these "good" continuations.

Finally, the authors provide a method - Alignment Resampling (AR), which is basically a best-of-n approach using an external reward model to score "n" generations from the policy and choosing the highest scoring ones. The authors provide a simple theoretical proof to justify their mitigation strategy and demonstrate experiments on the earlier benchmarks.

**Audience:**

Yes

**Audience Explanation:**

Yes, multiple LLM deployment personnel would be very interested in the findings of this paper. Essentially, the paper uses a reward model to select generations, which is simple and cost-effective.

**Broader Impact Concerns:**

None.

**Claims And Evidence:**

Yes

**Claims Explanation:**

C1. This paper's empirical work regarding watermarking on chosen alignment dimensions (truthfulness, safety, etc.) is definitely the first of its kind and serves as an important step towards a multi-faceted analysis. Relevant works like WaterJudge (Molenda et al.) only measure the "quality" or accuracy of such watermarked models - providing a single-phase analysis. This contribution is novel and well-grounded. In my opinion, this analysis is much more actionable than just "quality".

C2. The intuition of alignment and watermarking being orthogonal objectives looks correct based on the qualitative definitions presented. I am not sure if this is formal enough to be robust to all watermarking techniques, but for the scope of the paper, it is good enough, as the next section discusses mitigation strategies.

C3. Even though the strategy is straightforward, the results show that most of the misalignment produced by a policy is fixed with AR. More importantly, the detectability of the watermark remains more or less unchanged, which is desired.

**Requested Changes:**

**Major:**

M1. *Impact of different Reward Models*: I think this is the most critical point of improvement in the paper. The authors only experimented with Armo as the reward model (for all plots/experiments), which raises questions about the implicit biases built into the model and if the AR method is generalizable to ANY reward model. \
M2. Additionally, I am a bit concerned about the training phase of Armo. If the benchmarks have some dataset leakage with the training data of the reward model, it might be possible that AR is fundamentally compromised. I urge the authors not to just report the training configurations, but also the objectives and benchmark performances of Armo itself. \


**Minor:**
m1. "The Orthogonality Problem" example in 4.1 - I understand the spirit of the example presented here. However, the statement "The watermark thus systematically biases generation toward unsafe continuations" is too strong because, in certain cases, the continuations can be positively biased as well. How an average generation gets affected is demonstrated through its benchmark numbers, so I would soften the statement a bit. \
m2. For all benchmarks, GPT-4o-mini has been utilized as the evaluator. This can create a model-specific bias in evaluation and risks overfitting the mitigation strategy to the judge used for evaluations. Please compare against more evaluators
m3. Qualitative examples: I see no qualitative examples in the main text. I request authors to add more examples in the main text to let readers understand the diversity and quality of generations.

---

> ### Author Response · Authors · 2025-12-13
> **Response**
>
> We thank the reviewer for their insightful review. We share the reviewers opinion that the orthogonality argument is somewhat heuristic and appreciate the reviewers recognition of the novelty of our work. The AR technique is indeed BoN applied on watermarked outputs and our experiments show that this simple technique mitigates most of the alignment degradation without affecting the detectability of the watermark.
>
> Based on the review comments, we conducted several new experiments and updated the paper with the new results and figures. We have updated the pdf in the submission where the new text is marked in violet for ease of reviewing. This will be blended in if the paper is accepted. Please find our response to suggested changes below.
>
>
> **M1. _Impact of different Reward Models_:**
> This was raised by another reviewer and we agree with the viewpoint that similar results with additional reward model would help provide more concrete evidence for the efficacy of our proposed technique. To this end, we run additional experiment with the Skywork reward model (https://arxiv.org/abs/2410.18451). We choose this model since this is a competitive model on the reward bench leaderboard (https://huggingface.co/spaces/allenai/reward-bench). When we were writing the paper ArmoRM was among the top 10 on the leaderboard. These additional results are included in a new section in the appendix titled `Appendix K: Generalization to Alternative Reward Models`
>
> What we see in these results is that the technique generalizes to this new reward model. Across the 4 models (Phi-3-Mini, Mistral-7B, Qwen2-7B, and LLaMA-8B) we are able to recover truthfulmess, safety and decrease overrefusals for all but one model. For Phi-3-Mini there is a slight increase in overrefusals.
>
> **If the benchmarks have some dataset leakage with the training data of the reward model, it might be possible that AR is fundamentally compromised.**
> We appreciate the reviewer raising the concern about dataset leakage. We acknowledge that this is a pervasive problem in current research and merits some analysis. This prompted us to carefully audit the training data for the ArmoRM reward model. We collected all the datasets used in the ArmoRM training as listed in the ArmoRM paper (https://arxiv.org/abs/2406.12845). Of the ten dataset names collected we find two dataset names overlapping with our evaluation sets HH-RLHF (https://huggingface.co/datasets/Anthropic/hh-rlhf) and Beaver Tails (https://huggingface.co/datasets/PKU-Alignment/BeaverTails/viewer/default/30k_test). Both these datasets have a train and test split and we **were careful** to use the test split as the evaluation split and thus do not suffer from the dataset leakage issue. Based on this simple analysis we believe that the reward guided selection using the ArmoRM reward model in our experiments is not compromised.
>
>
> **Minor: m1. "The Orthogonality Problem" example in 4.1 - I understand the spirit of the example presented here. However, the statement "The watermark thus systematically biases generation toward unsafe continuations" is too strong**
> We soften the claim to `This mechanism can introduce systematic biases that may increase the likelihood of unsafe continuations, depending on the model.`
>
> **m2. For all benchmarks, GPT-4o-mini has been utilized as the evaluator.**
> We think the central goal of our paper is to study the effects of watermarking on alignment properties. While different LLM-As-A-Judge models can introduce some variance in the evaluation results, this line of investigation lies outside the current scope of our work. We will add this as future work in conclusion.
>
> **m3. Qualitative examples: I see no qualitative examples in the main text.**
> We have replaced Figure 1 in the paper with a better figure with more qualitative examples spawning all three alignment criterion.
>
>
> We appreciate the reviewers thoughtful engagement with our work and the valuable insights they have provided. We believe our systematic diagnosis of watermark-alignment interactions and the practical AR solution address an important gap in the field, particularly given the increasing deployment of watermarked LLMs. Please let us know if our response clarifies your questions.

---

### Decision · Action_Editor_bnzM · 2026-01-14

**Recommendation:** Accept as is

**Audience:**

Yes

**Audience Explanation:**

This paper is closely relevant to two popular sub-areas in LLM -- alignment and watermarking.

**Claims And Evidence:**

Yes

**Claims Explanation:**

All the reviewers commonly agreed to accept this paper. The claims are well supported by experiments, including the degration of alignment caused by watermarking and mitigation by Alignment Resampling, and the authors also provided a theoretical justification.